# A Language and Compiler View on Differentiable Programming

**Fei Wang, Tiark Rompf**
Department of Computer Science
Purdue University
West Lafayette, IN 47906, USA
{wang603,tiark}@purdue.edu

## Abstract

Current and emerging deep learning architectures call for an expressive high-level programming style with end-to-end differentiation and for a high-performance implementation at the same time. But the current generation of deep learning frameworks either limits expressiveness and ease of use for increased performance (e.g., TensorFlow) or vice versa (e.g., PyTorch). In this paper we demonstrate that a "best of both worlds" approach is possible, based on *multi-stage programming* and *delimited continuations*, two orthogonal ideas firmly rooted in programming languages research.

## 1 Introduction

Popular deep learning frameworks rely on computation graphs to enable efficient automatic computation of gradients through reverse-mode automatic differentiation (AD) (Speelpenning, 1980; Rumelhart et al., 1986). How exactly those computation graphs should be constructed and managed is a subject of active debate: systems in the style of Theano (Al-Rfou et al., 2016) and TensorFlow (Abadi et al., 2016) assemble *static* computation graphs in a rather explicit way, which are run after a distinct compilation step. Systems in the style of Torch (Collobert et al., 2011), including PyTorch (Paszke et al., 2017), use *dynamic* computation graphs, which only record a trace of operations for the backward pass as the program is executed.

The main benefit of the static approach is that it offers a larger surface for analysis and optimization. Much like in an aggressive whole-program compiler, high-level optimizations can be applied that make training and inference much more efficient. The apparent downside of the static approach, however, is the rather clunky programming interface offered by current frameworks, the absence of sophisticated control flow constructs, and the inability to use standard debugging facilities. This leads to a situation pointedly described as "I use PyTorch at home and TensorFlow at work"[1], where programmers tend to prefer systems like PyTorch with their simpler API but are sometimes forced to port their code to systems like TensorFlow for increased performance in deployment.

In this paper, we argue that these current deep learning frameworks and their design decisions are just points in a larger space of possible options, and that implementations that combine the benefits of both approaches are imminently possible. In particular, we draw attention to two key ideas from programming languages research – *multi-stage programming* and *delimited continuations* – which have the potential to put deep learning frameworks on a firmer and more principled basis.

This paper makes the following specific contributions:

(1) We frame the "graph construction" model of Theano and TensorFlow as an instance of *multi-stage programming* (*staging* for short), and argue that the explicit stage distinction between graph construction and execution makes it inherently preferable over PyTorch's "just a library" model, due to its greater optimization potential. We further show that implementation techniques for multi-stage programming exist that eliminate most or all of the perceived drawbacks of TensorFlow and similar systems.

---

[1]https://www.quora.com/Is-Pytorch-better-than-Tensorflow-for-general-use-cases-1/answer/Roman-Trusov

(2) We present a new model of reverse-mode AD that does not rely on a tape, computation graph, or other explicit data structure. Instead, it uses *delimited continuations* to inject backward computation into the normal program control flow. With proper language support, the implementation can rely exclusively on operator overloading and does not require code transformations as in Myia (Breuleux & van Merriënboer, 2017), Tangent (Wiltschko, 2017), or DLVM (Wei et al., 2017).

(3) We show that combining these orthogonal ideas, staging and reverse-mode AD via delimited continuations, leads to an efficient implementation of both forward and backward computation without any extra difficulty. The result is easily mapped to existing optimizing compiler pipelines for tensor computations.

## 2 THE ESSENCE OF DEEP LEARNING FRAMEWORKS

### 2.1 GRAPH CONSTRUCTION AS MULTI-STAGE PROGRAMMING

The idea of constructing static computation graphs is closely related to the idea of *staging* in programming languages. Already 30 years ago (Jørring & Scherlis, 1986), researchers observed that many computations can be naturally separated into stages delineated by frequency of execution or availability of data. Similar to computation graphs in TensorFlow, often the "way to compute" is given, yet the "data to compute" is not. The idea to treat staging as an explicit *programming model* was popularized, among others, by Taha & Sheard (2000). Since then, modern staging approaches have been proposed that blend normal program execution with delayed construction of an *intermediate program representation* (IR), which may be a computation graph or a more customary abstract syntax tree (AST). An example is the Lightweight Modular Staging (LMS) framework (Rompf & Odersky, 2010), which provides a rather seamless implementation of staging in the Scala language. As the following example shows, the only thing that distinguishes IR construction from normal computation is the addition of Rep types on staged function arguments:

```
def totalScore(names: Rep[Vector[String]]) = {          def nameScore(name: Rep[String]) = {
  val scores = for ((a,i) <- names.zipWithIndex) yield (i * nameScore(a))      name.map(c => c - 64).sum
  scores.sum                                             }
}
```

Here, a simple name score is computed for a vector of strings. The types Rep[Vector[String]] and Rep[String] are the only give-aways that an IR is constructed. The implementation crucially relies on type inference and advanced operator overloading capabilities, which extend to built-in control flow constructs like if, for, and while so that normal syntax can be used. Like in the example, an expression for ((a,i) <- names) yield ... becomes a series of method calls with closure arguments names.map((a,i) => ...). In Python, decorators can achieve something similar.

Once the IR graph has been built, it can readily be mapped to standard tensor compiler pipelines (e.g. XLA, NNVM) or to purpose-built compiler frameworks that directly extend LMS (e.g. Delite and OptiML (Sujeeth et al., 2014; 2011)). Since staging in LMS is driven by the Scala type system, staging can be turned on and off by parameterizing code over Rep vs. normal types (Ofenbeck et al., 2017). In this way, the *same* source code can be used with normal execution for debugging, and with high-performance compilation in production, eliminating the need for different frameworks.

### 2.2 DELIMITED CONTINUATIONS FOR REVERSE-MODE AD

To set the stage, we recap the well-known implementation of forward mode AD, assuming that the model is a function $\mathbb{R} \to \mathbb{R}$. As shown in Figure 1a, class NumF has a field $x$ (the value) and a field $d$ (the tangent). The arithmetic operators are overloaded (see def *) to compute both the primal value and tangent for intermediate NumFs. The code is concise but not efficient for functions $\mathbb{R}^n \to \mathbb{R}$.

We present a first version of our new reverse-mode AD in Figure 1b. Keeping the straight-forward coding fashion of the forward mode, our implementation switches to *continuation-passing style* (CPS): the overloaded operators take a *delimited continuation* as parameter – a callback that embodies remaining computation – and construct the intermediate NumR $y$ with adjoint initialized as $0$. A continuation can be thought of as the return address of a function, but in our case, the *delimited* continuation $k$ acts more like a normal function call than a jump to a return address: invoking $k$ will return control to the caller after execution, so that the program can proceed at the call site afterwards. Here, the continuation $k$ is supposed to carry on with the forward propagation, set the final result's

```
class NumF(val x: Double, val d: Double) {   class NumR(val x: Double, var d: Double) {   class Num(val x: Double, var d: Double) {
  def *(that: NumF) =                           def *(that: NumR) = { (k: NumR=>Unit) =>     def *(that:Num) = shift {(k:Num=>Unit)=>
    new NumF(x * that.x,                          val y = new NumR(x * that.x, 0.0); k(y)      val y = new Num(x * that.x, 0.0); k(y)
      this.d * that.x + that.d * this.x)          this.d += that.x * y.d                       this.d += that.x * y.d
  ...                                             that.d += this.x * y.d                       that.d += this.x * y.d
}                                               }                                            }
                                                ...                                          ...
                                              }                                            }
a + a * a                                     (a * a)(aa => (aa + a)(aaa => aaa.d = 1.0))  reset { (a + a * a).d = 1.0 }
```

Figure 1: Automatic Differentiation in Scala: (a) forward mode, (b) reverse mode in continuation-passing style (CPS), (c) reverse mode using delimited continuations, with shift/reset operators.

adjoint to 1, and propagate backward while updating adjoints. After the call $k(y)$ returns, the code updates the adjoints of this and that based on $y.d$.

However, using NumR in Figure 1b is cumbersome. For a simple model of $a + a * a$, the continuations have to be explicitly constructed as closure expressions (see Figure 1b bottom). Fortunately, *delimited control operators* exist that enable programming with delimited continuations in direct style, without making the continuations explicit everywhere. The shift and reset operators (Danvy & Filinski, 1990) work together to capture a partial return path up to a programmer-defined bound: in our case the remainder of the forward pass. In Figure 1c, the keyword shift provides access to a delimited continuation that reaches up to the nearest enclosing reset further up the call chain. The Scala compiler transforms all the code inbetween accordingly (Rompf et al., 2009). The implementation of Num is almost identical to NumR (modulo added shift), and the computation model is just $a + a * a$ enclosed in reset with a final $.d = 1$ operation. Of course, this can be further encapsulated.

With delimited continuations, we successfully contained the reverse AD logic inside the overloaded operators, no more complicated than in forward mode. In comparison, the model of Pearlmutter & Siskind (2008) relies on a non-local code transform that builds explicit closures for the backward pass, and also requires additional care to support nested functions. Our implementation is so concise that it can serve as a *specification* of reverse mode AD and be used to teach AD to students.

## 2.3 COMBINING STAGING AND CONTINUATIONS

We now combine staging and reverse AD. The result is a highly efficient implementation that generates flat code without involving extra data structures. We label the fields of Num as Rep[Double], as in Figure 2a. This means that the generated code will be free of Num objects and only manipulate Double values directly. LMS permits easy use of control flow in static computation graphs, and generating code from a program that uses shift and reset will result in appropriately CPS-transformed code. In particular, the while-loop in Figure 2 will be staged as a recursive function, storing intermediate values on the call stack. It is worth noting that after staging by LMS, the generated code is free of front-end complications. We obtain a flat representation of combined forward and backward computation that can be optimized together. The example is easily extended from arithmetic on Doubles to linear algebra on tensors by using Rep[Vector[Double]] instead of Rep[Double] in class Num.

```
class Num(val x: Rep[Double], var d: Rep[Double])   void squash_loop(double temp_x, double &temp_d) {
{...}                                                 if (temp_x > 1.0) {
def squash(x: Num) = {                                  double y_d = 0.0;
  // Divide by 2.0 until less than 1.0                  squash_loop(0.5 * temp_x, y_d); // recursive call, update y_d
  var temp = x                                          temp_d += 0.5 * y_d;
  while (temp > 1.0) temp = 0.5 * temp                } else
  temp                                                  temp_d += 1.0;
}                                                    }
```

Figure 2: Lightweight Modular Staging (LMS): (a) Num implementation and example usage (while loop), (b) generated C code for gradient computation

## 3 CONCLUSIONS

We have shown how two ideas from programming languages research – *multi-stage programming* and *delimited continuations* – can put deep learning systems on a more principled basis, eliminate unnecessary accidental trade-offs, and lead to concise yet efficient implementations.

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
