# OpenReview forum: "A Language and Compiler View on Differentiable Programming"
_ICLR.cc/2018/Workshop — Accept_

### Official Review · AnonReviewer1 · 2018-03-08

**Rating:** 6
**Confidence:** 1

**Review:**

This paper propose a proof of concept for for automatic differentiation capable language that while it’s easy to use like languages with dynamic compilation graphs, has the advantage of static graph languages such as graph optimizations which affects performance.

Designing an intuitive and easy to use programming language for machine learning that has good performance is an active area of research and any potential breakthrough could have a big impact. Unfortunately I could not quite evaluate this paper due to reasons mentioned below:

Cons:
Unfortunately it’s heavily tailored toward programming languages researchers with lots of references to terminology and concept that might not be familiar to machine learning researchers which are not defined in text.
The paper shows proof of concept but have not developed the method.
Assume readers familiarity with Scala syntax

Pros:
Seems easy modification on top of existing systems, if one is willing to pay the cost of switching to Scala.

---

### Official Review · AnonReviewer3 · 2018-03-09

**Rating:** 7
**Confidence:** 2

**Review:**

The paper makes three main claims: 1) the authors show that the “graph construction” model used by certain deep learning frameworks can be viewed as an instance of multi-stage programming, 2) they propose a new model of reverse-mode automatic differentiation based on delimited continuations, and 3) they show that these two ideas can be combined into a framework which retains the ease of use of a define-by-run framework like PyTorch while offering the analysis and optimization capabilities of a define-and-run framework like TensorFlow.

I do think that work on making deep learning frameworks both efficient and easy to use is very relevant to the ICLR community, however having only a passing familiarity with programming language theory, I find myself lacking the required background material to properly evaluate this submission and can only offer an outsider’s perspective.

Can the authors elaborate on the difference between the reverse mode AD model they propose and the one proposed by Pearlmutter & Sisking, as well as what they mean by the reliance on non-local code transform attributed to the Pearlmutter & Sisking model?

Also, am I correct in asserting that source code transformation is in general sufficient to allow for the usability of frameworks like PyTorch and the efficiency of frameworks like TensorFlow (which a framework like Myia should, in theory, allow for), and that the main contribution of the paper is in the form of the specific AD implementation the authors propose?

I am giving this submission an accepting score but am not very confident in my evaluation; for my final score I intend to rely on the judgement of reviewers more familiar with the subject at hand.

(Very) minor nitpick: the Theano citation would probably look cleaner using the short BibTex entry provided by http://deeplearning.net/software/theano/citation.html.

---

### Official Review · AnonReviewer2 · 2018-03-10
**Interesting perspective about bridging NNs and FP, few evidences in real case**

**Rating:** 6
**Confidence:** 4

**Review:**

The paper describes the construction method of the neural network with reverse-mode automatic differentiation using functional programming, especially reset/shift operators, and the combination of those results and the multi-stage programming paradigms.
The perspective of using delimited continuation shown in the paper is interesting, which successfully constructs the forward/backward operation sequences without generating any additional structures about computation graphs.

There may exist a bit room for improvement of description about the design of the Num class in Fig. 1. The current paper describes the operator (e.g. the "*") as a sequence of calculations; generating the object representing the return value, invoking the continuation k, and performing the gradient propagation respectively. These are all necessary to realize the correct calculation, but are still not essential for (only looking at the perspective of) constructing the neural networks. It might be better to further separate the actual forward/backward operations from the "common procedure" of each operator.

In most cases, the majority of actual time consumption of neural networks is usually caused by functions with a large time complexity (e.g., matrix multiplication and convolution). Although the better compiler/runtime pipelines may generate/run more efficient programs than the naive calculation, I have no idea that "how much efficiently" these processes work. Could you show some results using the proposed paradigms in actual networks?

---

### Decision · Program_Chairs · 2018-03-20
**ICLR 2018 Workshop Acceptance Decision**

**Decision:**

Accept

**Comment:**

Congratulations, your paper was accepted to the ICLR workshop.